# Durvalumab and guadecitabine in advanced clear cell renal cell carcinoma: results from the phase Ib/II study BTCRC-GU16-043

Yousef Zakharia [1] ✉, Eric A. Singer[2,3], Satwik Acharyya[4], Rohan Garje[1], Monika Joshi[5], David Peace [6], Veera Baladandayuthapani[4], Annesha Majumdar[7], Xiong Li[4], Claudia Lalancette [8], Ilona Kryczek [4], Weiping Zou [4] & Ajjai Alva[4]

Epigenetic modulation is well established in hematologic malignancies but to a lesser degree in solid tumors. Here we report the results of a phase Ib/II study of guadecitabine and durvalumab in advanced clear cell renal cell carcinoma (ccRCC; NCT03308396). Patients received guadecitabine (starting at 60 mg/$m^2$ subcutaneously on days 1-5 with de-escalation to 45 mg/m2 in case of dose limiting toxicity) with durvalumab (1500 mg intravenously on day 8). The study enrolled 57 patients, 6 in phase Ib with safety being the primary objective and 51 in phase II, comprising 2 cohorts: 36 patients in Cohort 1 were treatment naive to checkpoint inhibitors (CPI) with 0-1 prior therapies and 15 patients in Cohort 2 were treated with up to two prior systemic therapies including one CPI. The combination of guadecitabine 45 mg/m2 with durvalumab 1500 mg was deemed safe. The primary objective of overall response rate (ORR) in cohort 1 was 22%. Sixteen patients (44%) experienced stable disease (SD). Secondary objectives included overall survival (OS), duration of response, progression-free survival (PFS), clinical benefit rate, and safety as well as ORR for Cohort 2. Median PFS for cohort 1 and cohort 2 were 14.26 and 3.91 months respectively. Median OS was not reached. In cohort 2, one patient achieved a partial response and 60% achieved SD. Asymptomatic neutropenia was the most common adverse event. Even though the trial did not meet the primary objective in cohort 1, the tolerability and PFS signal in CPI naive patients are worth further investigation.

Immune checkpoint inhibitors (CPI) directed against PD1/L1 are standard first-line therapy in advanced ccRCC in combination with either anti-CTLA4 antibodies or with small molecule vascular endothelial growth factor receptor (VEGFR) inhibitors[1,2]. However, not all patients benefit from these regimens, and treatment-related toxicities can be a limiting factor. Hence, there is a need to improve the safety and efficacy of anti-PD1/L1 therapy through novel rational combinations to reverse and prevent tumor immune evasion.

Epigenetic changes, including DNA hypermethylation, are associated with tumor progression[3]. Higher average methylation rates are associated with higher RCC stage and grade and carry a poorer prognosis[4]. The chemokines CXCL9, CXCL10, and CXCL11 in the tumor

[1]University of Iowa Holden Comprehensive Cancer Center, Iowa City, IA, USA. [2]Ohio State University Comprehensive Cancer Center, Columbus, OH, USA. [3]Rutgers Cancer Institute of New Jersey, New Brunswick, NJ, USA. [4]University of Michigan, Ann Arbor, MI, USA. [5]Penn State Cancer Institute, Hershey, PA, USA. [6]University of Illinois at Chicago, Chicago, IL, USA. [7]Big Ten Cancer Research Consortium, Indianapolis, IN, USA. [8]BRCF Epigenomics Core University of Michigan, Ann Arbor, MI, USA. ✉e-mail: yousef-zakharia@uiowa.edu

micro-environment are chemo-attractants for activated NK and Th1 cells and mediate antitumor immunity[5,6]. Preclinical data suggest hypermethylation-induced silencing of TH1-type chemokines signaling results in tumor immune evasion in RCC cell lines (A-498, HTB-46, and CRL-1611)[7]. Treatment with decitabine, a hypomethylating agent (HMA), in these cell lines increased CXCL9/10 levels. In mouse xenograft models, combination therapy with an HMA and CPI led to higher levels of CXCL9/10, reversal of immune evasion, and potent tumor regression. Furthermore, HMAs have been shown to enhance tumor antigen expression and endogenous antigen processing, increase the expression of MHC and co-stimulatory molecules, and boost effector T-cell function[8,9].

Guadecitabine is a hypomethylating agent shown to induce a dose-dependent decrease of global DNA and gene-specific methylation in preclinical models[10]. It is resistant to degradation by cytidine deaminase, resulting in a longer half-life and longer exposure window for its active metabolite, decitabine[11]. Durvalumab, an anti-PD-L1 antibody is approved for advanced urothelial and lung cancers[12].

Here, we show the combination of guadecitabine 45 mg/m$^2$ with standard dose and schedule of durvalumab is safe with promising efficacy signal that is worth further evaluation, especially in CPI treatment naive patients.

## Results

### Patient characteristics

Fifty-seven patients with advanced ccRCC were enrolled across 5 academic centers within the Big Ten Cancer Research Consortium.

Six patients (3 at each dose level of guadecitabine 60 mg/m$^2$ and 45 mg/m$^2$) were enrolled in phase Ib and 51 patients in phase II. These doses were chosen based on prior clinical trials in hematologic malignancies[11]. Thirty- six patients were CPI naive (Cohort 1), and 15 patients were CPI refractory (Cohort 2). Baseline characteristics of patients enrolled are summarized in Table 1. In cohort 1, the median age was 68 years (range, 40–85). Most patients (69%) were male, seven (19%) had mixed histology, and 15 (42%) had received one prior systemic therapy mainly VEGFR inhibitor 11/15 (73%). Thirty-four patients (94%) had intermediate International Metastatic RCC Database Consortium (IMDC) risk, 2 patients (6%) had poor IMDC risk stratification, and 6 (17%) had sarcomatoid histology. In cohort 2, all patients received prior CPI (7 patients treated with nivolumab single agent, 5 patients with ipilimumab and nivolumab, 3 patients treated with pembrolizumab and axitinib), 13 patients (87%) had intermediate risk per IMDC risk stratification, and 2 (13%) had poor-risk. Four patients (27%) had sarcomatoid histology, and 5 (33%) had mixed histology.

### Efficacy

At data cutoff, median follow-up was 20 months, and one subject was not evaluable for efficacy. Response assessment was performed using RECIST 1.1, (progressive disease (PD), stable disease (SD), and partial or complete response (PR/ CR)). The primary endpoint of investigator-assessed ORR in cohort 1 was 22% (8/36), one subject (2.7%) achieved CR and 7 (20%) achieved PR. Sixteen patients (44%) experienced a best response of stable disease (SD) lasting longer than 6 months, providing a disease control rate (DCR = CR + PR + SD > 6 months) of 66%.

Secondary endpoint included ORR in cohort 2; the addition of guadecitabine resulted in a PR in 7% (1/15) and 60% (9/15) had SD. By immune-related Response Criteria (irRC), in cohort 1 the immune-related ORR was 22% (8/36) and DCR 61%. Supplementary Table 1.

Other secondary endpoint included median time to response of 1.8 months (1.6–2.1) and median duration of response of 7 months (3.2-15.6). Many of the responses were durable as illustrated in Fig. 1a. In cohort 1 (CPI naive), the median PFS was 14.26 months (7.13- 24.5) as opposed to 3.91 months (3.09- 9.1) in cohort 2 (CPI refractory) (Fig. 1b). Median treatment time was 9.035 months (0.23 – 28.98), and 3.22 months (0.23 – 17.45) in cohort 1 and 2 respectively.

**Table 1 | Patient characteristics**

|  | Phase Ib (N = 6) | Phase II, cohort 1 (N = 36) | Phase II, cohort 2 (N = 15) |
|---|---|---|---|
| Sex |  |  |  |
| Female | 1 (17%) | 11 (31%) | 4 (27%) |
| Male | 5 (83%) | 25 (69%) | 11 (73%) |
| Age, median (range) | 63 (50–72) | 68 (40–85) | 65 (46–93) |
| Race |  |  |  |
| White | 5 (83%) | 33 (92%) | 15 (100%) |
| Black or African American | 1 (16.7%) | 1 (2.8%) | 0 (0%) |
| Asian | 0 (0%) | 1 (2.8%) | 0 (0%) |
| Unknown | 0 (0%) | 1 (2.8%) | 0 (0%) |
| Ethnicity |  |  |  |
| Hispanic or Latino | 0 (0%) | 0 (0%) | 1 (6.7%) |
| Non-Hispanic | 6 (100%) | 35 (97%) | 14 (93%) |
| Unknown | 0 (0%) | 1 (2.8%) | 0 (0%) |
| IMDC risk |  |  |  |
| Poor | 0 (0%) | 2 (6%) | 2 (13%) |
| Intermediate | 6 (100%) | 34 (94%) | 13 (87%) |
| Favorable | 0 (0%) | 0 (0%) | 0 (0%) |
| Clear cell renal cell carcinoma |  |  |  |
| Mixed | 2 (33%) | 7 (19%) | 5 (33%) |
| Pure | 4 (67%) | 29 (81%) | 10 (67%) |
| Sarcomatoid histology |  |  |  |
| Yes | 2 (33%) | 6 (17%) | 4 (27%) |
| No | 3 (50%) | 23 (64%) | 9 (60%) |
| NA | 1 (17%) | 7 (19%) | 2 (13%) |
| ECOG performance status |  |  |  |
| 0 | 4 (67%) | 20 (56%) | 6 (40%) |
| 1 | 2 (33%) | 16 (44%) | 9 (60%) |
| Any prior therapy | 6 (100%) | 15 (42%) | 15 (100%) |

IMDC International Metastatic RCC Database Consortium

Median OS was not reached in either cohort. The 1- and 2-year OS for cohort 1 was 92.9% and 85%, respectively and for cohort 2 was 78% and 62.4%, respectively (Fig. 1c).

### Safety and adverse events (AEs)

In the dose de-escalation phase, grade 3/4 neutropenia in 2 out of 3 patients was deemed a DLT with guadecitabine 60 mg/m$^2$. Since phase Ib of the trial was enrolling amid the peak of COVID-19 pandemic, and the risk of neutropenia, infection, and hospitalization were of major concerns, the decision for dose de-escalation of guadecitabine to 45 mg/m$^2$ was made and deemed the RP2D. There were no DLTs with guadecitabine at 45 mg/m$^2$.

Adverse events were reported by the treating investigators and were graded per Common Terminology Criteria for Adverse Events (CTCAE; version 4.03).

In the intention-to-treat (ITT) population, the incidence of treatment-related adverse events (TRAEs) of any grade with guadecitabine was 24.4%, durvalumab was 20.1%, and either guadecitabine or durvalumab was 35.4%. Neutropenia (54.4%) was the most frequent AE from guadecitabine (G3/4: 39%), and lipase elevation (19.3%) was the most common AE from durvalumab (G3/4: 10.5%). Cutaneous AEs were the most common immune-mediated AEs (35%), other AEs, which were generally mild (all ≤Grade 3), included thyroid dysfunction, diarrhea, dyspnea, pneumonitis, myalgia, and hepatotoxicity. No treatment-related deaths were reported. Tables 2 and 3.

Dose delays due to adverse events (AEs) occurred in 34 (60%) patients; of those, 4 came off treatment due to AEs (two had

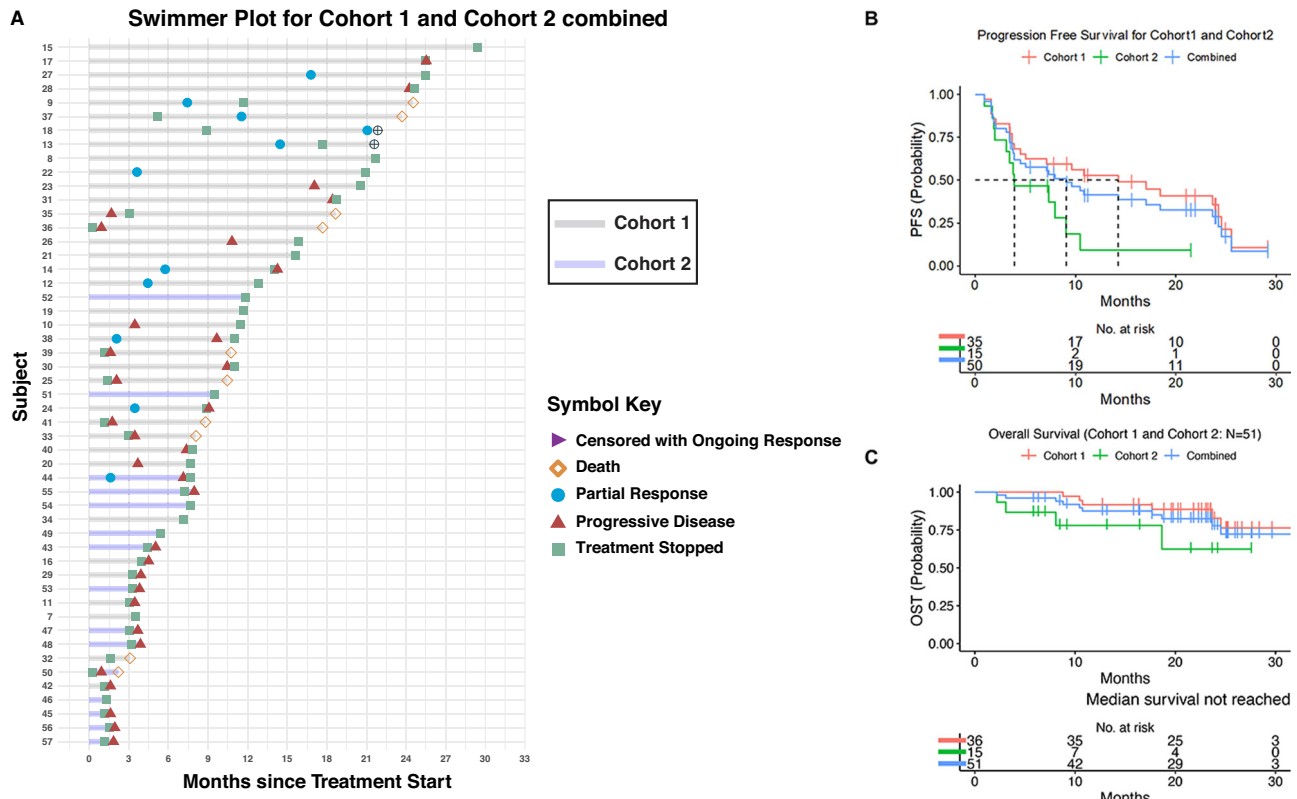

**Fig. 1 | Efficacy outcomes in the treatment of patients with mRCC using dur-valumab and guadecitabine. A** Swimmer plot showing treatment response (RECIST 1.1) and survival characteristics for Cohort 1 ($n = 36$ gray) and Cohort 2 ($n = 15$ violet). Symbols indicate time point when progressive disease (triangles) or partial response (circle) was achieved. Arrowheads indicate patients whose stable disease or response was ongoing at the time of data analysis. **B** Progression-free survival and **C** overall survival based on cohort, Kaplan-Meier plot. Source data are provided as a Source Data file.

pneumonitis, one had prolonged grade 2 lipase levels, and one had gross hematuria). Nine patients (16%) required oral corticosteroids for the treatment of immune-related adverse events (IRAEs).

### Effect of different immune cell populations on response

Predefined analysis of basic leukocyte subsets was performed using peripheral blood collected before treatment (C1D1) (Supplementary Fig. 1A). Most leukocyte subsets at baseline show similar distribution across response categories (Supplementary Fig. 1B–F). However, myeloid-derived suppressor cells (MDSC) were inversely associated with response, showing the highest levels in PD and the lowest levels in PR (Fig. 2A).

Preclinical data suggest that the presence of IFNγ in the periphery may be an important factor in response. We therefore measured the IFNγ expression in circulating effector T cells and $CD7^+CD3^-$ innate lymphoid cells (ILCs) at baseline. Patients responding to treatment had the highest expression of IFNγ in $CD8^+$ T cells (Fig. 2B, C), but not in other analyzed cell subsets (Supplementary Fig. 1I, J). TNFα was highly expressed in responders compared to non-responders in $CD8^+$ T cells (Fig. 2D, E), CD4 + T cells (Fig. 2F), and ILCs (Fig. 2G). Next, we examined the pro-inflammatory cytokines IL-17 and IL-22. The levels of inflammatory cytokines were only elevated in the $CD8^+$ T cells (Fig. 2H), though this subset is not the major producer of these cytokines; IL-17 (Supplementary Fig. 1N, O) being expressed at higher levels by $CD4^+$ T cells and ILCs.

The T-cell effector response is modulated at the transcriptional level by major transcriptional factors (TFs). To investigate the possi-bility of transcriptional reprogramming of T cells, we tested the kinetics of changes in the presence of T-bet, GATA3, RORγt, and Foxp3. Most TFs are expressed at comparable levels in lymphoid cells at baseline (Supplementary Fig. 1P–R). However, we found that Foxp3 expression in $CD4^+$ T cells (Fig. 2K, L) is significantly lower, and RORγt in $CD8^+$ T cells (Fig. 2 I, J) is significantly higher in the PR group com-pared to PD and consistent with the higher levels of IL-17 (Fig. 2H) expressed by CD8 cells in the responders' group).

### Biomarker association with immune-mediated toxicity

We performed a post-hoc analysis of the possible association of the peripheral leukocyte phenotype with immune-mediated toxicity. We categorized patients based on their CTCAE grade of immune-mediated toxicity into Grade 1–2 (mild), Grade 3–5 (severe), and no immune-mediated toxicity (none). We found no significant differences in the major leukocyte subtypes between patients in these categories. Also, the production of cytokines by T cells at baseline level showed no significant correlation with the toxicity categories. Only the level of IL-2 produced by ILCs (Fig. 3A), but not by T cells (Supplementary Fig. 2A, B), were associated with toxicity. Interestingly, IL-2 and IFNγ were induced by immunotherapy in CD4 and CD8 T cells, respectively only in patients with severe immune-mediated toxicity (Fig. 3B–E).

We also looked at the relationship between transcriptional factors T-cell expression and immune-mediated toxicity. Baseline levels of T-bet, driver of the Th1/cytotoxic response, showed an inverse associa-tion with immune-mediated toxicity (Fig. 3F–H), but was induced with treatment only in patients who manifested severe toxicity (Fig. 3I).

### Methylation modulation

Pre-planned measurement of methylation at LINE-1 (long interspersed nuclear elements) repetitive regions was conducted by pyrosequen-cing to verify the effect of guadecitabine, using genomics DNA isolated from peripheral white blood cells collected at C1D1 and C2D8

**Table 2 | All emergent adverse events profile in intention-to-treat (ITT) population**

| Adverse Events (AE), in alphabetical order | Guadacitabine (G) and Durvalumab (D) (ITT population n = 57) | |
|---|---|---|
| | **Any grade n (%)** | **G3/4 n (%)** |
| Abdominal pain | 14 (24.6) | 1 (1.8) |
| Anemia | 16 (28.1) | 1 (1.8) |
| Anorexia | 15 (26.3) | 1 (1.8) |
| Constipation | 11 (19.3) | |
| Cough | 14 (24.6) | |
| Diarrhea | 13 (22.8) | |
| Dyspnea | 16 (28.1) | 2 (3.5) |
| Extremities edema | 12 (21.1) | |
| Fatigue | 29 (50.9) | |
| Headache | 14 (24.6) | |
| Hyperkalemia | 12 (21.1) | |
| Hypertension | 14 (24.6) | |
| Hypothyroidism | 7 (12.3) | |
| Injection site reaction | 19 (33.3) | |
| Increased creatinine | 12 (21.1) | |
| Lipase Increased | 11 (19.3) | 6 (10.5) |
| Lymphopenia | 14 (24.6) | 3 (5.3) |
| Nausea | 16 (28.1) | 1 (1.8) |
| Neutropenia | 31 (54.4) | 22 (38.6) |
| Pain in extremity | 12 (21.1) | |
| Pruritus | 12 (21.1) | 1 (1.8) |
| Rash maculo-papular | 8 (14.0) | |
| Vomiting | 6 (10.5) | |
| Weight gain | 8 (14.0) | |
| White blood cell decreased | 25 (43.9) | 5 (8.8) |

**Table 3 | Immune-related adverse events (irAE) profile in intention-to-treat (ITT) population**

| Immune-related adverse events (irAE), in alphabetical order | Guadacitabine (G) and durvalumab (D), ITT population (n = 57) | |
|---|---|---|
| | **Any grade n (%)** | **G3/4 n (%)** |
| Arthralgia | 4 (7) | |
| Arthritis | | 3 (5) |
| Diarrhea | 13 (22.8) | |
| Dyspnea | 16 (28.1) | 2 (3.5) |
| Hypothyroidism | 7 (12.3) | |
| Lipase increased | 11 (19.3) | 6 (10.5) |
| Myalgia | 3 (5) | |
| Pneumonitis | | 3 (5) |
| Pruritus | 12 (21.1) | 1 (1.8) |
| Rash, maculo-papular | 8 (14.0) | |
| Increased AST | 3 (5) | |
| Increased ALT | 3 (5) | |

*AST* aspartate aminotransferase, *ALT* alanine aminotransferase

$P = 0.00007$, $P = 0.00001$, and $P = 0.000007$, for CXCL9, CXCL10, and CXCL11 respectively).

Next, we evaluated the changes in Th1 chemokines serum levels with response. We created the increased group based on more than 15% change and the decreased group based on less than 15% change at C2D8 compared to the baseline.

For this purpose we selected patients with substantial changes of more than 15% in chemokine level at C2D8 compared to the baseline (Fig. 5B–D), and grouped best responses as SD, PR, and CR together vs PD. Based on the binomial proportion test, the increase in all CXCL9, CXCL10, and CXCL11 was observed with the best RECIST response of (SD, PR, and CR) group ($P = 0.054$ and $P = 0.000208$, and $0.000208$ respectively) Supplementary Table 2.

There was a trend for improvement in PFS with the increase of CXCL9/10/11, an observation that needs further investigation in larger studies (Supplementary Fig. 4).

## Discussion

Epigenetic modulation has demonstrated clinical benefit and been utilized extensively in hematologic malignancies; however, its use in solid tumors is being explored increasingly. The two main classes of medications manipulating epigenetics include HMAs (e.g., decitabine, guadecitabine, azacytidine) and histone deacetylase inhibitors like panobinostat. Multiple preclinical studies have suggested the synergistic effect of epigenetic modulators and immunotherapy. In one example, treatment with HMAs led to up-regulation of cancer antigen expression and resulted in the simultaneous release of T-cell effector pro-inflammatory cytokines leading to T-cell-mediated tumor killing[13,14]. Decitabine was also found to enhance CD8[+] T-cell activation, proliferation, and cytolytic activity, which correlated with improved antitumor responses and survival of patients with solid tumors[15]. Recent work combining guadecitabine (30 mg/m²) and pembrolizumab in 34 patients with refractory solid tumors, reported similar constellation of side effects with neutropenia being the most common adverse AE, with DCR of 37%. Patients with clinical benefit had high baseline inflammatory signature on RNAseq and increases in intra-tumoral effector T cells[16]. Another study combining guadecitabine (45 mg/m²) and atezolizumab; in metastatic urothelial carcinoma after progression on prior CPI; was prematurely terminated for futility after enrolling 21 total patients with best response of SD in 4 patients. Biomarker work suggested the lack of DNA demethylation in tumors after 2 cycles of treatment. However, increased peripheral immune

timepoints. Samples for both timepoints were available for 25 patients. Figure 4A shows the change in methylation averaged across 3 CpG sites. A decrease in methylation was observed in 22/25 (88%) patients between C1D1 and C2D8 timepoints ($P = 1.165 \times 10^6$), showing overall the efficacy of guadecitabine.

We also conducted a post-hoc evaluation of the methylation level in a region of the CXCL10 promoter for 12 selected patients (top 15% and lower 15% changes in CXCL10). (Fig. 4B) shows the change in methylation averaged across 7 CpG sites. An overall decrease in methylation was observed in selected patients between C1D1 and C2D8 timepoints ($P = 0.0216$), suggesting an effect of guadecitabine. Of the 12 patients tested, 3 had a slight increase in methylation in the region of the CXCL10. There was good correlation for the change in methylation between LINE-1 and CXCL10 (Pearson = 0.852) (Supplementary Fig. 3). However, there was low correlation between change in methylation at the CpG sites tested for CXCL10 and change in protein expression measured by ELISA (Pearson = 0.244).

### Assessment of chemokines

Preclinical studies with the combination of hypomethylating agents and CPI had shown an association of elevated levels of the MIG/ CXCL9 and IP-10/ CXCL10 with response.

We performed a predefined analysis of the chemokine's distribution prior to and during the treatment. Plasma samples at baseline and after cycle 2 day 8 (C2D8 = 5 weeks) were analyzed for chemokines level by Luminex (Panel HCYP3MAG-63K-08 and for CXCL9, CXCL10 and CXCL 11 Millipore). Only Th1 chemokines (CXCL9, CXCL10, and CXCL11) were significantly induced by immunotherapy (Fig. 5A,

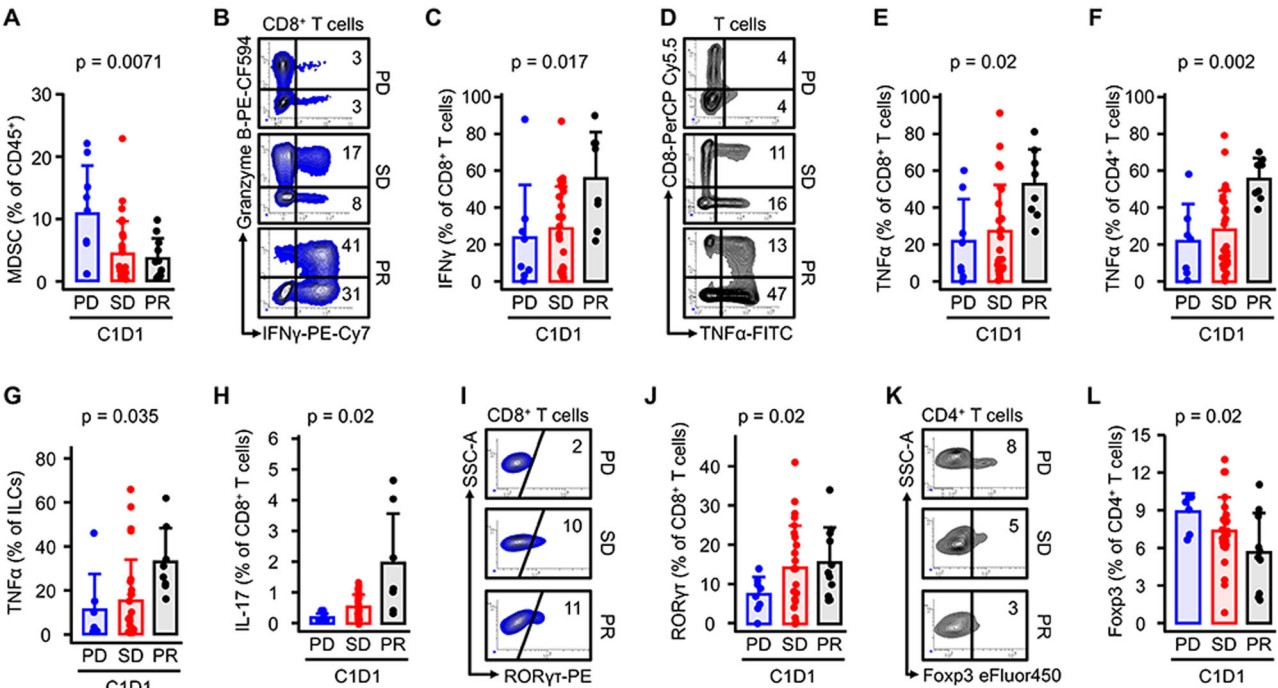

**Fig. 2 | Relationship between immune cell phenotype and response.** Mononuclear cells isolated from peripheral blood collected before treatment (C1D1) were analyzed by flow cytometry. Response assessment was performed using RECIST 1.1 and allowed patients to be grouped as those with progressive disease (PD *n* = 9), stable disease (SD *n* = 28), and partial or complete response (PR *n* = 11). **A** Myeloid-derived suppressor cells (MDSC) (*n* = 47) were gated as CD45+Lin (−) CD33+CD14− and CD45+Lin(−) CD33+HLA-DR−). Mean + SEM, two-way ANOVA. **B**−**H** Association

peripheral T-cell effector status with patients' responses. Percentages of IFNγ (*n* = 43) (**B**, **C**) TNFα (*n* = 42) (**D**−**G**), RORγt (*n* = 45) (**I**, **J**) and Foxp3 (*n* = 43) (**K**, **L**) in CD8+ or CD4+ T cells were analyzed by FACS. Results are expressed as mean + SEM, Kruskal-Wallis ANOVA. Gating started is shown in supplementary Fig. 1A. One representative dot plot from each of the response types is shown. Source data are provided as a Source Data file.

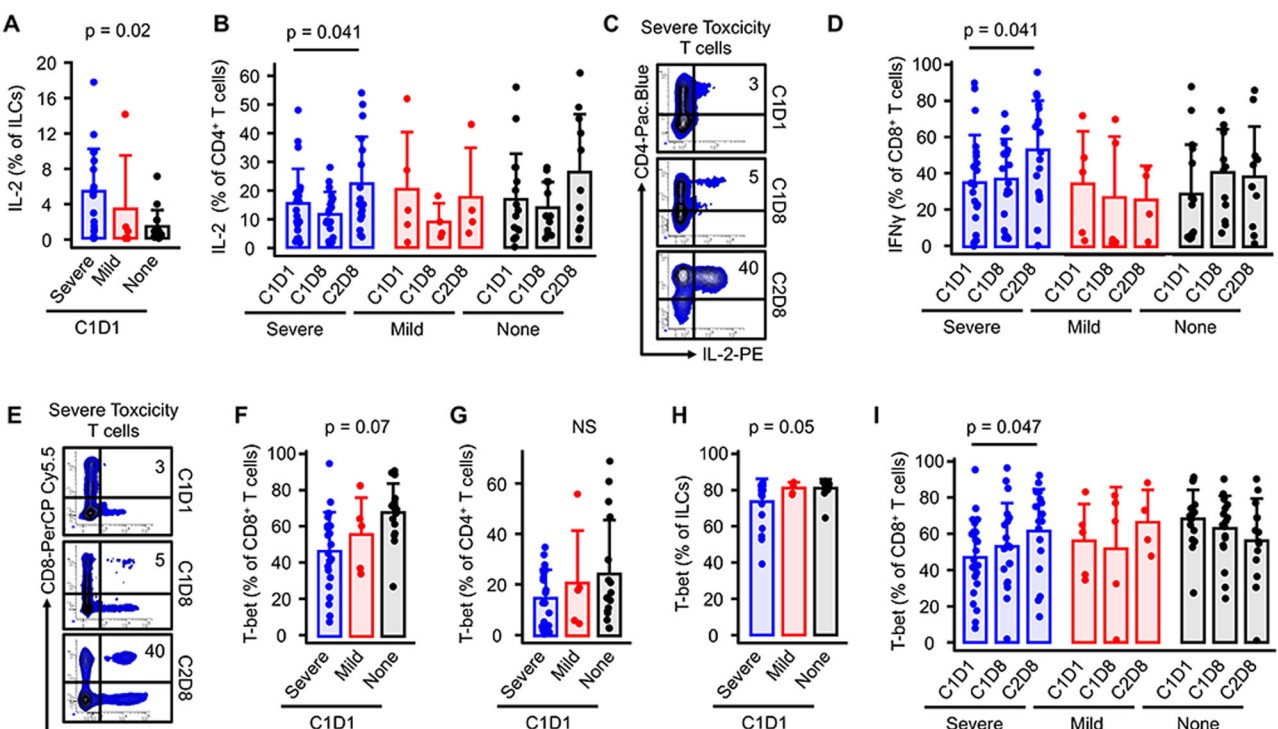

**Fig. 3 | Relationship between immune cell phenotype and immune-mediated toxicity.** Mononuclear cells isolated from peripheral blood collected before treatment (C1D1) and at 5 weeks (C2D8) were analyzed by flow cytometry. Patients were categorized based on their CTCAE grade of immune-mediated toxicity into

Grade 1−2 (mild) (*n* = 5), Grade 3−5 (severe) (*n* = 24), and no immune-mediated toxicity (none) (*n* = 19). Results are expressed as mean + SEM, one-sided paired *t* test. Gating started is shown in supplementary Fig. 1A. One representative dot plot from each time point is shown. Source data are provided as a Source Data file.

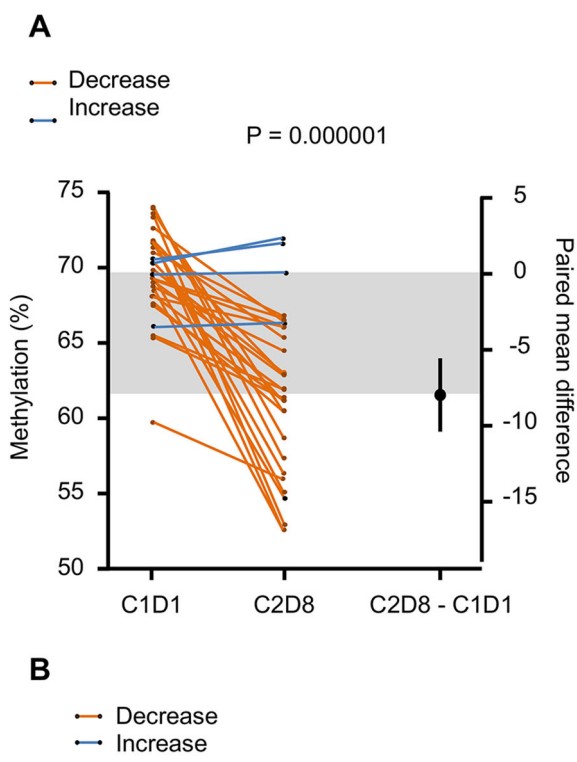

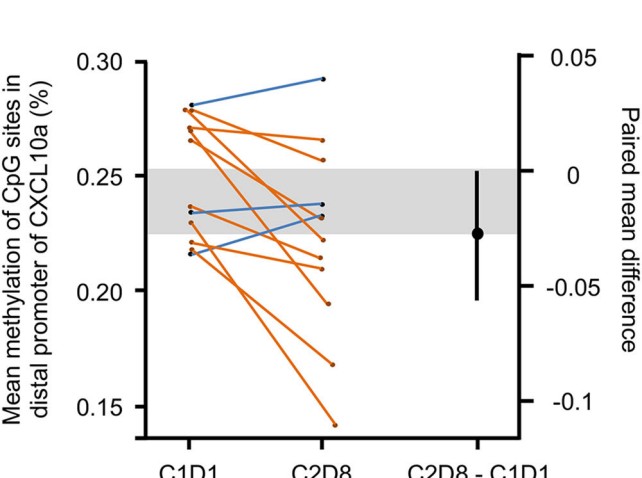

**Fig. 4 | Methylation changes with treatment. A** Methylation at LINE-1 repetitive element regions, a decrease in methylation was observed in 22/25 patients between C1D1 and C2D8 timepoints. **B** Gartner-Aldman plot of change in CXCL10 methylation between first and second endpoints (n = 12). One-sided paired *t* test. Source data are provided as a source data file.

activation and immune infiltration in tumors after treatment correlated with progression-free survival and SD[17].

To our knowledge, this study is one of few evaluating the safety and efficacy of a HMA with a checkpoint inhibitor in advanced solid tumors, specifically in RCC. The dosing of guadecitabine at 45 mg/m$^2$ subcutaneously on days 1–5 along with durvalumab (1500 mg IV on day 8) every 28-day cycle is deemed safe for phase II evaluation.

In the CPI naive subjects (cohort 1), the ORR with the combination therapy was 22%, which is lower than our predefined primary efficacy endpoint of an ORR of 45%. Forty-four percent achieved stable disease that lasted ≥6 months accounting for a disease control rate (DCR) of 66%. Patients had either IMDC intermediate- or poor-risk disease, which might explain the observed lower-than-expected response rate. The median PFS of 14.26 months in Cohort 1 (with 42% of patients receiving prior line of therapy) is intriguing and raises the question of

whether ORR is the optimal primary endpoint in metastatic RCC clinical trials where many tumors manifest response by central necrosis without accompanying shrinkage of 30% to meet the ORR cutoff per RECIST 1.1 criteria. We believe the observed median PFS in Cohort 1 deserves further validation in a randomized clinical trial. In the CPI refractory cohort, SD was reported in 60% (9/15), which is interesting despite the limited number of patients in this cohort. This observation might support the immunomodulatory effect of hypomethylating agents in reversing CPI resistance and is worth further evaluation.

At the time of data cutoff, median overall survival was not reached. It is noteworthy that the 1- and 2-year OS for cohort 1 of 92.9% and 85%, respectively, is encouraging in this high-risk patient population.

The safety profile of guadecitabine was consistent with other reported studies in hematological malignancies with the most common adverse event of neutropenia[18]. This AE was manageable with dose delay and dose reduction of guadecitabine. Two patients received filgrastim for neutropenia. There were no grade 5 events. The most common immune-related adverse events were dyspnea, diarrhea, elevated lipase, pruritus, and skin rash comparable to other first-line CPI-based therapies[1].

Flow cytometry on peripheral blood collected before treatment (C1D1) demonstrated that MDSC were inversely associated with response with the lowest levels in responders, suggesting that immune suppression driven by myeloid cells at baseline may influence patients' clinical response. Consistent with their suppressive nature, we found that Foxp3 expression in CD4$^+$ T cells is significantly lower in the PR group. Interestingly, RORγt-expressing CD8$^+$ T cells are significantly higher in responding patients compare to patients with progressive disease. CD8$^+$ T cells expressing RORγt (so-called TC17 cells) have been shown to promote inflammation, contribute to defense against infections, and participate in autoimmunity[19]. The impact of RORγt on TC17 is unclear, however, we have shown that RORγt activates stemness pathways in T cells inducing persistence and polyfunctionality[20]. This property may be critically important for controlling also CD8 T cell's response to neoantigens[21]. As expected, patients responding to treatment had the highest expression of IFNγ in CD8$^+$ T cells at baseline but not in other circulating lymphocytes subsets like CD4$^+$ T cells and ILCs. This is of interest given the involvement of IFNγ in host-tumor interactions[22] and because loss of the IFNγ signaling pathway impairs T-cell responses, permits tumor growth[23] and could be a mechanism of primary resistance to CPI[24].

Previous reports have suggested expansion of a large number of clonal CD8 + T cells preceding the development of grade 2–3 toxicity in the setting of CPI treatment[25]. Our data did not suggest significant differences in the major leukocyte subtypes between patients according to the severity of side effects. Baseline level of IL-2 produced by ILCs but not by T cells seem to correlate with toxicity. These data suggest that therapy-induced rather than baseline cytokine status of T cells determine toxic side effects.

Increased expression of chemokines such as CXCL9 and CXCL10 has been correlated with favorable prognosis in RCC[26]. Previous studies have associated the use of CPI with concomitant increase in both serum and tissue chemokines[27]. We observed a significant increase in serum CXCL9/10 with the study combination, and this increase seems to correlate with better clinical outcome. This correlation needs to be validated in larger clinical trials and could serve as a biomarker to predict responders from non-responders in CPI-based treatment.

LINE-1 refers to repetitive elements of DNA forming around 17% of the genome and can be a surrogate of global DNA methylation[28]. Utilizing serial blood samples from 25 patients at C1D1 and C2D8, we observed a decrease in methylation in 22/25 (88%) patients (P = 1.165 × 10$^6$), providing a proof-of-mechanism. In the aforementioned clinical trial combining guadecitabine and pembrolizumab in refractory solid tumors, a significant reduction in LINE-1 DNA

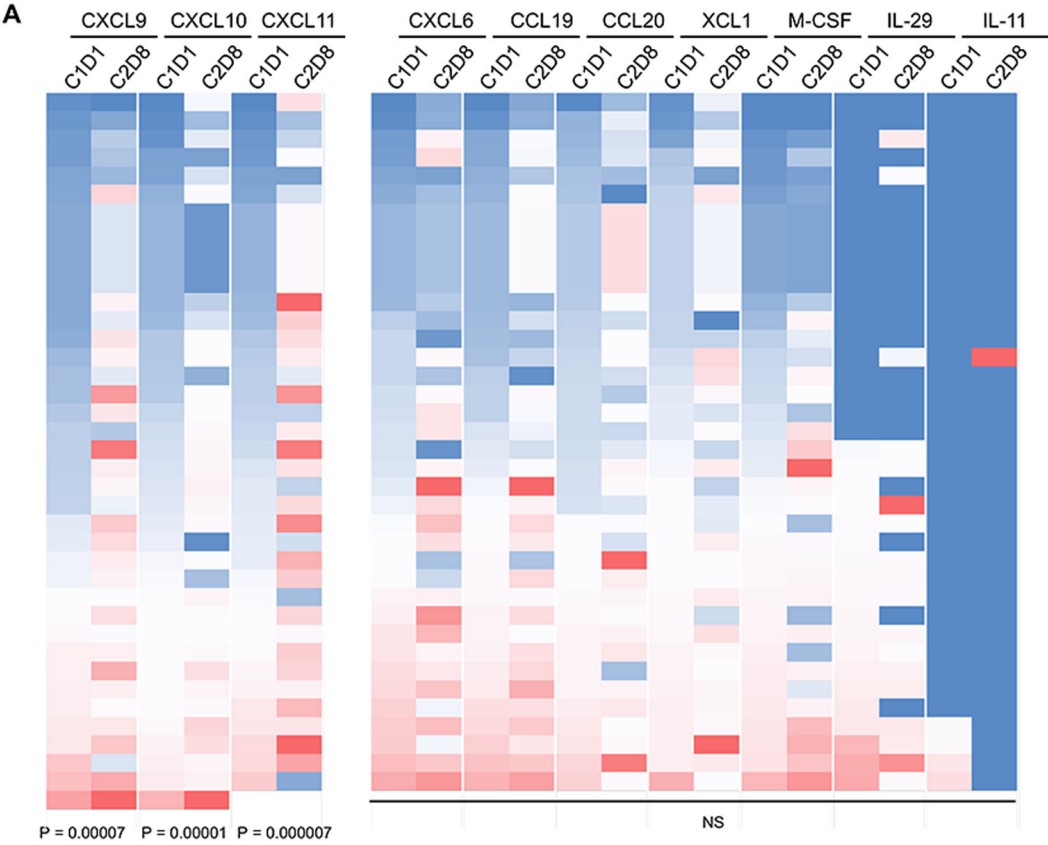

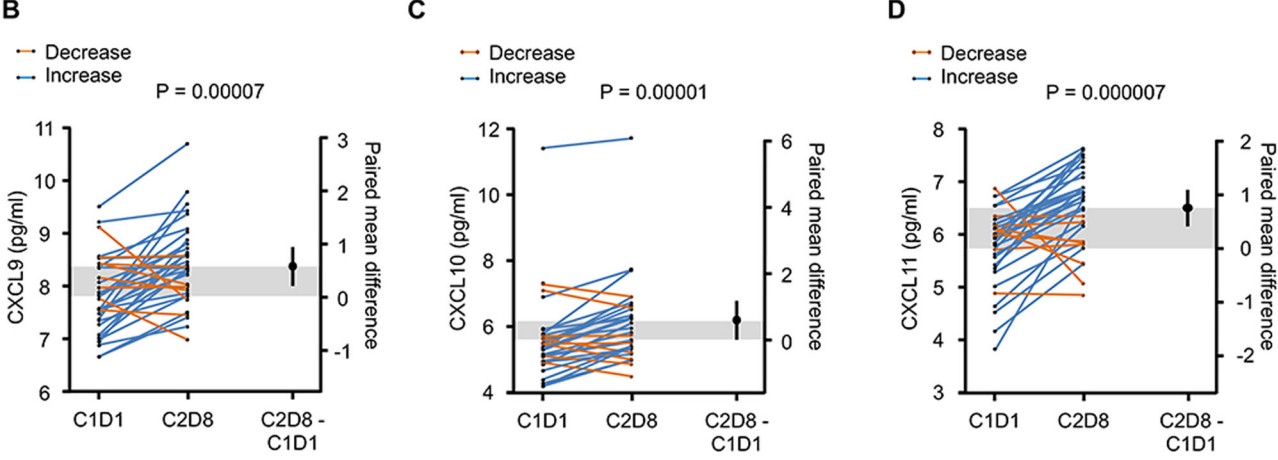

**Fig. 5 | The association between chemokine serum levels and patient clinical outcome.** Plasma samples from 34 patients at baseline prior to treatment (C1D1) and at 5 weeks (C2D8) were analyzed for presence chemokines by Luminex (Panel HCYP3MAG-63K-08 for CXCL9 and CXCL10, Millipore). **A** Heatmap generated from non-transformed data separate for each chemokine. Units are in pg/mL; one-sided paired *t* test. **B**–**D** Gardner-Altman plot with denotes log-transformed (**B**) CXCL9 (*n* = 34), (**C**) CXCL10 (*n* = 34), and (**D**) CXCL11 (*n* = 34) values at C1D1 and C2D8 collection timepoints. One-sided paired *t* test. Increased group is created based on >15% change, and decreased group is based on <15% change. The 15% cutoff was determined arbitrarily by testing significance in 15% increments of 15%, 30%, and 45%, respectively. Source data are provided as a Source Data file.

methylation following treatment, mostly pronounced in PBMCs samples at C2D8 (median 48.7%) compared with baseline (median 64.3%)[16]. It is worth noting that the dose of guadecitabine used in this trial was 30 mg/m², day 1–4 which might explain the differences noted in methylation reduction in comparison to our study.

Our study had several limitations, including small sample size in both cohorts, single-arm design, and relatively slow accrual. Although we had initially intended to conduct treatment research biopsies to enhance our biomarker analysis and gain a deeper understanding of

the tumor's underlying biology, the implementation of COVID-related restrictions in hospitals prevented us from carrying out this aspect of the study.

In conclusion, the combination of durvalumab and guadecitabine has an acceptable toxicity profile and promising activity especially in CPI-naïve patients with advanced ccRCC as first and subsequent lines of therapy. It could be further tested in patient populations where TKI and IO are not well tolerated and can cause early treatment discontinuation.

## Methods

### Study design and participants

BTCRC-GU16-043 (NCT03308396) was a multicenter, single-arm phase Ib/II clinical trial in patients with metastatic ccRCC (pure or mixed histology) conducted throughout the Big Ten Cancer Research Consortium (Big Ten CRC). Participants providing informed consent were enrolled between January 18, 2018, and September 24, 2020. All participating sites obtained approval from their respective Institutional Review Boards, and the study complied with all relevant regulations (Supplementary Table 3). The phase Ib portion evaluated the de-escalating doses of guadecitabine (level 0: 60 mg/m$^2$ and level −1: 45 mg/m$^2$ subcutaneously on Days 1–5) along with standard dose of durvalumab 1500 mg/m$^2$ on Day 8 every cycle (4 weeks). The phase II part included patients in two cohorts: Cohort 1 ($n = 36$) enrolled patients who were treatment naive or with one prior line of therapy that is not CPI. Cohort 2 ($n = 15$) enrolled patients with up to two prior systemic therapies including at least one CPI. All phase II patients were treated with the RP2D of guadecitabine along with durvalumab until disease progression or unacceptable toxicity or patient's withdrawal of informed consent. Treatment beyond first progression was permitted if patient was clinically stable and derived benefit with the study as per treating physician's discretion and provided additional informed consent.

Sex and/or gender were not considered in the trial design as no sex differences have been seen in previous clinical trials in ccRCC. The current trial recruited any patient independent of sex or gender. The sex of the participants was based on subject's self-report. Sex/gender was only reported in the subjects' demographics, and no gender analysis was carried out given the lack of evidence to correlate sex or gender with treatment outcome in metastatic ccRCC.

Eligible subjects were ≥18 years old with histologically confirmed metastatic ccRCC (pure or mixed) with at least one RECIST 1.1 measurable lesion. Other inclusion criteria were ECOG performance status of 0 or 1, life expectancy of ≥12 weeks, adequate bone marrow function including white blood cells (WBC) ≥ 3000/mm$^3$, absolute neutrophil count (ANC) ≥ 1.5 K/mm$^3$, hemoglobin ≥9 g/dL and platelet count ≥100,000/mm$^3$), adequate renal function (calculated creatinine clearance ≥40 cc/min using the Cockcroft-Gault formula) and adequate hepatic function (total bilirubin ≤ 1.5 × upper limit of normal (ULN), alanine aminotransferase (ALT) and aspartate aminotransferase (AST) ≤ 2.5× ULN). Key exclusion criteria included concurrent active infection requiring systemic therapy, ongoing another primary active malignancy within the last 5 years, active inflammatory or autoimmune disease, and history of allogenic organ transplant. Patients with CNS metastases were only allowed if CNS metastases were adequately treated with imaging stability for ≥4 weeks. The detailed inclusion and exclusion criteria are listed in the study protocol (available in the Supplementary Information). All participating subjects provided written informed consent; no compensation was offered to study subjects.

### Objectives and assessments

The primary objective of the phase Ib portion was to determine safety based on DLTs and to identify the RP2D of guadecitabine in combination with durvalumab. The primary objective of phase II was to assess the efficacy in cohort 1 (CPI naive) as defined by objective response rate (ORR = CR + PR) measured per RECIST v1.1 criteria.

The secondary objectives included 2-year overall survival, 12-month progression-free survival (PFS), duration of response (DOR), and ORR in cohort 2 (CPI refractory). Exploratory objectives included assessing the correlation of baseline and treatment changes in the levels CXCL9 and CXCL10 in serum with clinical response. Additionally, we evaluated the immunologic parameters such as CD3$^+$/CD8$^+$ tumor-infiltrating lymphocytes (TILs) of archival tumor tissue, PDL1 expression in tumor cells and TILs, change in LINE-1 demethylation with therapy, immune cell subsets in the serum and tissue, tumor

mutational burden, and epigenetic changes at baseline and in correlation with the best clinical response as per RECIST 1.1.

Disease assessment with axial imaging of chest, abdomen, and pelvis (CT/MRI) was done at screening and every 8 weeks thereafter until progression. The response was assessed per RECIST v1.1 criteria for the whole study. Response by immune-related RECIST criteria was evaluated as a secondary endpoint[29,30]. Toxicities were graded per Common Terminology Criteria for Adverse Events (CTCAE; version 4.03).

### Cell preparation and flow cytometry

PBMCs were isolated by density centrifugation using Lymphoprep (StemCell Technology) according to the manufacturer's protocol. Surface staining was performed for 30 min at 4 °C in MACS buffer (buffer (PBS, 2% FCS, 1 mM EDTA). For analysis of the expression of transcription factors and cytokines, surface-stained cells were fixed and permeabilized with the Perm/Fix Buffer Set (ThermoFisher) according to the manufacturer's instructions. For intracellular cytokine staining, PBMC were ex vivo restimulated with a leukocyte stimulation cocktail containing ionomycin and PMA (Sigma) for 4 h in the presence of brefeldin A and monomycin (BD Biosystem). Cells were acquired by Fortessa (BD Biosciences). Data were analyzed using DIVA software (BD Biosciences). All antibodies used for staining are listed in Supplementary Table 4 and were used at the recommended by manufacturer volume per test (ranging between 5–20 µl).

### Chemokine detection

For multiplex quantitative chemokine analysis the Luminex detection system was used. Serum samples were tested with two commercially available panels: Human Cytokine/Chemokine Magnetic Bead Panel III (HCYP3MAG-63K-08) and Human Cytokine/Chemokine/Growth Factor Panel A Magnetic Bead Panel (HCYTA-60K).

Samples were processed according to the manufacturer's instruction. Serum concentrations are given as mean in pg/ml with a 95% CI (Confidence interval). Values below the detection limit were interpreted as 0 of chemokine.

### LINE-1 methylation

Cell pellets were brought to the BRCF Epigenomics Core for DNA isolation and processing for measurement of LINE-1 methylation by pyrosequencing. Genomic DNA was extracted using Qiagen's All Prep DNA/RNA kit according to the instruction manual for the DNA extraction protocol. Genomic DNA was quantitated using Qubit BR dsDNA kit, and quality was assessed using TapeStation 4200 Genomic DNA kit. Five hundred nanograms of bisulfite were converted using the EZ-96 DNA Methylation Kit (Zymo Research, Irvine, CA). Bisulfite-converted DNA was amplified using LINE-1 primers and HotstarTaq Master Mix (Qiagen, Valencia, CA). LINE-1 primer sequences are from forward primer sequence: 5′-TTGAGTTAGGTGTGGGATATAGTT-3′, LINE-1 reverse primer sequence: 5′-[biotin]-CAAAAAATCAAAAAATT CCCTTTCC-3′, LINE-1 sequencing primer: 5′-AGGTGTGGGATATAGT-3′. This primer pair amplifies a region with 4 CpG sites. DNA methylation was measured using the Pyromark Q96 ID pyrosequencer (Qiagen, Valencia, CA). In all the samples, the 4th CpG site failed quality control after pyrosequencing. The first 3 CpG sites were therefore used for analysis.

### CXCL10 methylation

Amplicon bisulfite sequencing was used to measure methylation at 7 CpG sites in the promoter of the CXCL10 gene. Primers were designed using the EpiDesigner tool from Agena Biosciences (www.epidesigner.com). Genomic DNA was bisulfite converted, and amplicons were amplified using Qiagen HotStar polymerase. For each patient, amplicons were pooled at equimolar concentration before preparing libraries using the KAPA Hyper Prep library kit.

Final libraries were quantified using the KAPA Library prep quantification kit, pooled at equimolar concentration, and sequenced on a MiSeq Nano V2 flow cell with PE-150 cycles. The data generated was trimmed of adapter sequences, mapped to the human genome (hg38) using Bismark, and methylation data was extracted using MethylDackel. The resulting files contain methylation data as beta values for each CpG site/sample. These were imported in R for analysis.

### Statistical analysis
Using binominal test and with an assumed ORR of 25% with durvalumab monotherapy, 36 patients were required to detect an ORR of 45% with the combination of guadecitabine and durvalumab with 80% power assuming a one-sided 5% type I error with an exact binomial test. Progression-free survival (PFS) and Overall Survival (OS) were assessed with Kaplan–Meier survival analysis with associated 95% confidence intervals. The proportion of patients with each grade of adverse events as defined by CTCAE (version 4) was computed along with the 95% CI and reported in a tabular and descriptive manner. All analyses were computed using R version 4.02.

Since all immunological variables were continuous and had skewed observations, Kruskal–Wallis ANOVA or one-sided paired $t$ test was used for assessing significance. $P < 0.05$ was considered statistically significant.

### Study oversight
The original protocol and all amendments were approved by the relevant institutional review board (IRB). The study was conducted in accordance with the protocol, Good Clinical Practice guidelines, and the provisions of the Declaration of Helsinki. All patients provided written informed consent. A data and safety monitoring committee oversaw the study. All data were collected by investigators and associated site personnel, analyzed by statistician, and interpreted by the authors. All authors participated in reviewing and editing the manuscript, approved the submitted draft, had full access to the data used to write the manuscript, vouched for their accuracy, and attested that the study was conducted in accordance with the protocol.

### Reporting summary
Further information on research design is available in the Nature Portfolio Reporting Summary linked to this article.

## Data availability
The study protocol is provided in the Supplementary Information file. The authors, sponsors, and the Big Ten Cancer Research Consortium are committed to providing scientific researchers access to anonymized data from clinical trials and associated biomarkers for the purpose of conducting legitimate scientific research. Authors and sponsors are obligated to protect the rights and privacy of trial participants and, as such, will evaluate and fulfill requests for sharing clinical trial data with qualified external scientific researchers. Interested investigators can obtain and certify the data transfer agreement (DTA) and submit requests to the corresponding author. Proposals will be vetted by the Big Ten Cancer Research Consortium and participating institutions. Investigators and institutions who consent to the terms of the DTA form, including but not limited to the use of these data for the purpose of a specific project and only for research purposes, and to protect the confidentiality of the data and limit the possibility of identification of participants in any way whatsoever for the duration of the agreement, will be granted access for 6 months. All requests will be evaluated within 8 weeks. The remaining data are available within the Article, Supplementary Information, or Source Data file. Source data are provided with this paper.

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

## Acknowledgements

We thank AstraZeneca Pharmaceuticals, LP (ESR-16-12275) and Astex Pharmaceuticals, Inc. (EP23) for supporting the trial. Big Ten CRC Administrative Headquarters at Hoosier Cancer Research Network, Inc., Inc., Indianapolis, IN, for providing trial management. The authors thank Kristina Greiner for her editing assistance.

## Author contributions

All authors (Y.Z., E.S., S.A., R.G., M.J., D.P., V.B., A.M., X.L., C.L., I.K., W.Z., A.A.) had access to primary clinical trial data and vouch for the accuracy and completeness of the data and for the fidelity of the trial to the protocol. All authors (named above) contributed substantially to the concept and design of the manuscript, data analysis and interpretation, and writing of the manuscript. All authors approved the final version of the manuscript.

## Competing interests

Y.Z.: advisory boards: Bristol Myers Squibb, Amgen, Roche Diagnostics, Novartis, Janssen, Eisai, Exelixis, Castle Bioscience, Genzyme Corporation, Astrazeneca, Array, Bayer, Pfizer, Clovis, EMD Serono, Myovant. Grant/research support from: institutional clinical trial support from New-LinkGenetics, Pfizer, Exelixis, Eisai. Data safety and monitoring committee: Janssen Research and Development. Consultant honorarium: Pfizer, Novartis. E.S.: institutional research funding: Medivation/Astellas. Advisory boards: Merck, Johnson & Johnson, Vyriad. Data safety monitoring board: Aura Biosciences. M.J.: institutional research funding: AstraZeneca. Advisory boards: Seagen, Gilead. Travel and accommodation: DAVA Oncology. Mentor on diversity grant: Bristol Myers Squibb. S.A., R.H., D.P., V.B., A.M., XL, C.L., I.K., W.Z., A.A.: declare no competing interest.
