## [Peer Review File · Nature Communications]

Durvalumab and Guadecitabine in Advanced Clear Cell Renal Cell Carcinoma: Results from the Phase Ib/II Study, BTCRC-GU16-043REVIEWER COMMENTS

Reviewer #1 (Remarks to the Author): with expertise in immunotherapy, epigenetics, clinical

In their manuscript titled "Phase Ib/II Study of Durvalumab and Guadecitabine in Advanced Clear Cell Renal Cell Carcinoma: Big Ten Cancer Research Consortium BTCRC-GU16-043", Zakharia and coll. report the clinical outcome of the study and the results of some translational studies performed in treated patients.

The study focuses on the highly relevant topic of therapy resistance in RCC patients and on the potential role of epigenetic drugs (specifically the hypomethylating agent guadecitabine), as a strategy to reverse such phenomenon once combined with immune checkpoint inhibitors (specifically the anti-PDL-1 durvalumab). In their study Authors divided enrolled patients into two separate Cohorts of ICI naive and ICI-resistant RCC patients, respectively. Primary objective of the study was Objective Response Rate (ORR) in patients from Cohort 1 that was not reached. Comprehensively, the data reported (and as reported) do not seem to support Author's conclusions in ICI naive RCC patients and do not provide sufficient evidence in ICI-resistant subjects.

Although the topic of epigenetic modeling is of emerging clinical interest in solid malignancies, the study as reported presents to this reviewer some major criticisms as reported below:

Major Comments

1) The primary objective was ORR in patients from Cohort 1 but Authors reported clinical results of this Cohort side by side to those of Cohort 2 (secondary objective) throughout the manuscript. This seems quite misleading based on the primary objective of the study, also in view of the characteristics of patients enrolled in the two Cohorts (e.g. ICI naive vs ICI resistant). Furthermore, the numerosity of the Cohorts is highly unbalanced (42 vs 15), as it is the number of previous lines of therapy. Along this line, 64% of patients enrolled in Cohort 1 were treatment naive. In addition, patients enrolled in the Phase I are described together with subjects enrolled in the Phase II and this does not help the reader to understand immediately the demographic of patients enrolled in the Phase I of the study.

2) Six patients were enrolled in the Phase I of the study. After the initial 3 patients treated per protocol at Dose level 0, the dose of guadecitabine was reduced at Level -1 "... to avoid dose delays for neutropenia and for DSMC recommendations...". This seems methodologically quite unusual and in contrast with the study protocol since only one patient experienced a DLT (grade 3 neutropenia). It would be useful for the readers to have explained how was the Level 0 dose of guadecitabine identified by the authors in this study.

3) Treatment exposure does not seem to be reported in the manuscript being quite important to evaluate feasibility, tolerability and efficacy of the therapeutic combination explored in the study.

4) The comparison of the PFS of this study with that of study CM025 should greatly softened; a similar

consideration applies to the comparison with standard first line treatment of RCC patients mentioned by the Authors.

5) Along the line of point 4 above, it would have been highly desirable from the Authors to discuss their results in light of the recently published clinical trials that have combined guadecitabine with diverse ICI in solid tumors. Readers would have had much better picture of what is ongoing in the field of epigenetic combinations.

6) Good to have exploratory endpoints in the study. However, Authors should make an effort to clarify several aspects of their work including the results obtained on “archival” tissues that do not seem to be reported in the manuscript. Furthermore, the strong demethylation of LINE-1 reported in the manuscript should be discussed in light of available literature from studies combining guadecitabine with ICI where the extent of demethylation of LINE-1, at similar doses of guadecitabine, was much lower than that observed in this study.

7) The upregulation of CXCL9 and CXCL10 expression is quite interesting and should be explored further in RCC patients treated with these combinations. However, including studies on gene promoters would greatly help the interpretation of these results, and the mechanistic role of guadecitabine in this phenomenon.

Reviewer #2 (Remarks to the Author): with expertise in biostatistics, clinical trial study design

Thank you for giving the opportunity to review your manuscript. Please find below my comments/questions:

1. Why only doses of guadecitabine were chosen? How long was the total duration of the treatment schedule? I might have missed it in the manuscript, if it's there.
2. How would you comment on the broad range age? Is this a limiting factor?
3. Which is the list of toxicities that you would deem as dose limiting and therefore count towards an event in Phase 1b? You only mention grade 3 neutropenia as the only one of interest.
4. There is no justification as to the method used for the dose finding approach. You base your decisions on 6 patients, 3 on each dose. This is a very small number to declare a RP2D, how confident are you on this result?
5. There is no description as to which summary statistics were used to describe data, no mention on the methodology or rules of the escalation/de-escalation study. In addition, there is no justification as to the sample size of Phase 1b.

6. It'd be helpful to see summaries of prior treatment in Cohort 2 (duration, type of treatment). How was this accounted for in the analysis?

Reviewer #3 (Remarks to the Author): with expertise in immunotherapy, renal cancer, clinical

Thank you for asking me to review the manuscript, "Phase Ib/II Study of Durvalumab and Guadecitabine in Advanced Clear Cell Renal Cell Carcinoma: Big Ten Cancer Research Consortium BTCRC-GU16-043". In this manuscript by Zakharia et al., the authors reported a safety and overall response data from a Phase Ib/II trial using durvalumab plus a hypomethylating agent (guadecitabine) in patients with metastatic clear cell RCC (mccRCC). They had 2 cohorts of patients. Cohort 1 included patients who were immune checkpoint therapy naïve (n=42) and cohort 2 included patients who were previously treated with immune checkpoint therapy (n=15). They reported that the combination is overall safe. In Phase II, an OR of 6% and PFS of 18.4 months were noted in cohort 1, while cohort 2 had an OR of 7% and PFS of 3.9 months. Further, they also reported immune monitoring studies on pre-treatment and post-treatment PBMC samples.

Overall, the manuscript has significant shortcomings both in the clinical and correlative analysis. My specific comments are as follows:

My specific comments are as follows:

Clinical:

1. Although, the combination is safe, OR and PFS rate was not impressive even in cohort I over the standard of care options. Therefore, utility of clinical translation is limited.
2. Most of the patients (93%) were intermediate or poor risk. One would have expected a better response to immune checkpoint therapy than demonstrated in the trial.
3. Demographics are not well matched. 77% of the patients were male. Age range is also very wide.
4. In cohort 2, it was not clear whether patients received single agent immune checkpoint therapy, or combination of two immune checkpoint agents or immune checkpoint agent plus TKI. This information is important.
5. Responses evaluation was not centralized but investigator assessed.
6. Statistical methodology, study design is not properly explained in the manuscript.

Immune monitoring data:

Authors had 4 main figures reporting the immune monitoring data. However, the data lacks rigor in investigating the immunological correlates. The authors have used PBMCs for assessing the immunological correlates. Though this technique is less invasive, yet the peripheral immune landscape

does not always reflect the intra-tumoral immune landscape. In that case, pre and post-treatment biopsies would provide critical relevant information.

1. The authors have mentioned differences in abundance of MDSCs. However, more information is required on the type of MDSCs. What markers were used to define them?
2. Authors have mentioned that TNFa was higher in all lymphocyte subsets (Figures 2D-G). But data is shown only for T cells and ILCs.
3. Rather than saying higher expression of FoxP3 in T cells, better to show the abundance of these cell types? Do T cell suppression assays to assess Treg phenotype?
4. RORgt in CD4 is TH17. What is its impact in CD8 T cells and their ability to mount anti-tumor immune response?
5. Authors have mentioned that certain immune parameters are “correlated” with response/toxicity. However no statistical tests have been done to measure correlation and no R values have been given.
6. Instead of looking only at cxcl9/10, maybe the authors can consider doing a luminex panel/s? Also flow is not the best way to look at chemokine expression.
7. Fig. 4c,d are main figures but there is no statistical analysis was reported.
8. Checking hypomethylation in specific genetic regions which have been mentioned in the manuscript via bisulfite genomic seq so that the change in methylation can be correlated with the immune parameters discussed (for eg, increase in cxcl9/10).
9. On what basis was the 15% cutoff selected for binomial gating of patient samples?
10. The authors have discussed that the increase in cxcl9/10 could serve as a biomarker to “predict” responders from non-responders? However, the authors have not shown any correlation of response with pre-treatment baseline values of cxcl9/10. Then can this be considered a “predictive” biomarker?
11. In all Figures, please draw lines to indicate which groups are being considered for statistical tests.
12. Include details of the statistical tests done to investigate differences in immunological correlates.
13. Is Data shown as Mean+/-SEM or SD?
14. All Fig. 2 panels have been wrongly referred to as Figure 1 in the legend.
15. In Fig. S1a- no live/dead gating.
16. In Fig. S1a- no specific markers (for eg. CD19) used to define B cells.
17. Fig. S1a- Specific markers for ILCs.
18. Table S2- CXCL9/10 changes were in the same patients or different patients?
19. Table S2- SD, PR and CR have been grouped together, instead group SD with PD?

Reviewer #4 (Remarks to the Author): with expertise in immunotherapy, renal cancer, clinical

Phase 1/2 study of PDL1i + hypomethylating agent in advanced RCC. The combination of Durvalumab + guadecitabine proved to be too toxic at starting dose, and the ph2 piece of the trial did not meet its efficacy goal. An in-depth peripheral-blood based biomarker analysis was conducted (mostly to address heterogeneity in peripheral immune composition / activation of various compartments; less so around the interplay between methylation and immune activation) and is included with several findings of

interest.

- Protocol states dose escalation plan (section 5.3 protocol): “• Six patients will be enrolled at dose level 0. If 2 or fewer patients experience a dose limiting toxicity, the study will continue to the phase II portion. • Alternately, if 3 or more patients have a dose limiting toxicity at dose level 0, 6 patients will be accrued at the lower dose (dose -1). If 2 or fewer patients experience a dose limiting toxicity, the study will continue to phase II at dose level -1.” The results section of the paper states, 3 + 3 patients were enrolled at 60mg (level 0) and 45mg/m² (level -1) with one 1 DLT seen. The protocol section 5.5 “Treatment plan for the Phase II Portion” states “three of five Phase Ib subjects who received dose of 60mg/m²...”

It is unclear how many pts were treated on ph1b portion (3 vs 5) and why it wasn't a total of 6 , per the protocol?

- Table 1 / results: further information should be provided regarding exposure to prior therapies (median # lines at the least) to help put findings in perspective

- The authors should clearly state which imaging exact assessment methodology was used on the trial – are responses reported here per RECIST 1.1, modified RECIST, iRECIST, etc.? “disease control rate” should be clearly defined (looks like they are proposing CR+PR+SD>6mo)

- It is not meaningful to report a median OS for cohort 1 (6 patients). Landmark OS more helpful; could consider reporting that for all patients – rather than separate for ph1b and ph2

- Adverse events: attributions were per treating investigators? Please state clearly; table 2: are these treatment-emergent or treatment-related?

- IRAE – acronym not defined in text

- Adverse events: the ph1b findings should be reported by dose level (separate table). With that, a more detailed explanation is needed for the DSMC's recommendation to pursue DL-1 as RP2D –Safety results section states ‘to avoid dose delays for neutropenia’; Table 2 notes G4 neutropenia in 39% of the cohort , but the results section does not highlight this appropriately (G4 not mentioned in the text at all; table 2 does not break things down by cohort). How soon did neutropenia set in? were patients dose-reduced/rechallenged? If yes, how did they do? all these would be relevant to better understand this key issue with the combination.

- legends figures 2 and 3 each contain an error (reference figure 1 and 2, respectively; when it should be 2 and 3, respectively).

- Flow cytometry (“Effect of different...”) text reference significant correlation (MDSC) or the lack thereof; p-values provided in figures 2 and 3. the text (and fig legend) should reference which statistical test was used to test for association (not just naked p value). I cannot find this in the methods section either.

- Biomarker findings: for the most part these are testing hypotheses around the IO piece of the regimen; inverse efficacy correlation with myeloid populations is not a novel finding, nor is the interferon-gamma signal

- Biomarkers and immune-mediated toxicity: the manuscript should provide a dedicated table breaking down incidence and types of irAEs (vs. table 2 which is all AEs)

- T-bet baseline / on treatment; this is an interesting finding; which cells were tested for expression? where there patients who did not express T-bet at baseline but did so at a later timepoint, and did they suffer high-grade AEs?

- CXCL9&10 levels in peripheral blood: this is one of the most relevant correlative analysis for the

combination, and it is interesting to see the trends highlighted in figure 4. what was the proportion of patients with increase vs. decrease? The text (and figures, ideally) should state. The No at risk #s in 4C suggest 7 increase 25 decrease? How many w/o change? Why are they not included in Fig 4C/D?

- Figures 4C and D: would be more interesting to see increase/decrease curves in 1 graph and visualize the comparison

- Pyrosequencing LINE-1: this is the 2nd biomarker analysis of specific interest for the COMBINATiON, i.e. of central importance to the paper. Considering this, the section is rather short (vs flowcytometry of relevance to durva which in turn is not novel and could be shorter...). How do changes in CpG methylation correlate with efficacy? with neutropenia? the authors should provide further detail.

- were genomics assessed and is there correlation between alteration in recurrent events (histone-regulating genes?) and response?

We would like to thank the reviewers for the great points raised about the strength and weakness of our submission, here we address all valid questions and concerns raised by the reviewers.

Thank you!

Authors,

Major Comments

1) The primary objective was ORR in patients from Cohort 1 but Authors reported clinical results of this Cohort side by side to those of Cohort 2 (secondary objective) throughout the manuscript. This seems quite misleading based on the primary objective of the study, also in view of the characteristics of patients enrolled in the two Cohorts (e.g. ICI naive vs ICI resistant). Furthermore, the numerosity of the Cohorts is highly unbalanced (42 vs 15), as it is the number of previous lines of therapy. Along this line, 64% of patients enrolled in Cohort 1 were treatment naive. In addition, patients enrolled in the Phase I are described together with subjects enrolled in the Phase II and this does not help the reader to understand immediately the demographic of patients enrolled in the Phase I of the study.

We thank you the reviewer for the valid points, this was clarified in the manuscript. We separated primary from secondary endpoints We also separated phase 1 from phase 2 cohort 1 hence the current phase 2 cohort 1 N is 36.

Cohort 1 and 2 are unbalanced per protocol statistical design

2) Six patients were enrolled in the Phase I of the study. After the initial 3 patients treated per protocol at Dose level 0, the dose of guadecitabine was reduced at Level -1 "... to avoid dose delays for neutropenia and for DSMC recommendations...". This seems methodologically quite unusual and in contrast with the study protocol since only one patient experienced a DLT (grade 3 neutropenia). It would be useful for the readers to have explained how was the Level 0 dose of guadecitabine identified by the authors in this study.

This was modified in the study, Dose level 0 of Guadecitabine SC at 60 mg/m² Dailyx5 was tested in hematology clinical trials as single agent and was deemed safe. Issa et al: Lancet Oncol. 2015 Sep;16(9):1099-1110

Same dose was utilized in 2 other phase 3 trials in hematologic malignancies: ClinicalTrials.gov Identifier: NCT02920008 and NCT02907359

3) Treatment exposure does not seem to be reported in the manuscript being quite important to evaluate feasibility, tolerability and efficacy of the therapeutic combination explored in the study.

Median treatment time for subjects in cohort 1 is 9.03 months (0.23 – 28.98 months), and in cohort 2 is 3.22 months (0.23 – 17.45 months). This was updated in the manuscript.

4) The comparison of the PFS of this study with that of study CM025 should greatly softened; a similar consideration applies to the comparison with standard first line treatment of RCC patients mentioned by the Authors.

This was modified in the discussion part of the paper. We deleted the paragraph comparing PFS to CM025. The modified paragraph reads as following:

The median PFS of 14.26 months in Cohort 1 (with 36% of patients receiving prior line of therapy) is intriguing and raises the question of whether ORR is the optimal primary endpoint in metastatic RCC clinical trials where many tumors manifest response by central necrosis without accompanying shrinkage of 30% to meet the ORR cutoff per RECIST 1.1 criteria. We believe the observed median PFS in cohort 1 deserves further validation in a randomized clinical trial.

5) Along the line of point 4 above, it would have been highly desirable from the Authors to discuss their results in light of the recently published clinical trials that have combined guadecitabine with diverse ICI in solid tumors. Readers would have had much better picture of what is ongoing in the field of epigenetic combinations.

We thank the reviewer for the comment, recent publication combining guadecitabine and pembrolizumab was added in the paper along with reference, Papadatos-Pastos D *et al*: Journal for ImmunoTherapy of Cancer 2022;10:e004495. doi:10.1136/jitc-2022-004495.

Recent work combining guadecitabine (30 mg/m²) and pembrolizumab in 34 patients with refractory solid tumors, reported similar constellation of side effects with neutropenia being the most common adverse AE, with DCR of 37%. Patients with clinical benefit had high baseline inflammatory signature on RNAseq and Increases in intra-tumoural effector T-cells.

6) Good to have exploratory endpoints in the study. However, Authors should make an effort to clarify several aspects of their work including the results obtained on “archival” tissues that do not seem to be reported in the manuscript. Furthermore, the strong demethylation of LINE-1 reported in the manuscript should be discussed in light of available literature from studies combining guadecitabine with ICI where the extent of demethylation of LINE-1, at similar doses of guadecitabine, was much lower than that observed in this study.

We had an initial plan to perform biomarkers on matched biopsies and compare to archival tissue, however due to the COVID-19 pandemic and the participating universities policies restricting research activities, we could not perform research biopsies. Hence biomarkers were performed on blood samples only.

We added the supportive literature to the discussion section from Papadatos-Pastos et al JITC paper 2022, although it is worth noting that the dose of guadecitabine used is 30 mg/m², day 1-4 which might explain the differences noted in methylation reduction.

Added to discussion: LINE-1 refers to repetitive elements of DNA forming around 17% of the genome and can be a surrogate of global DNA methylation. Utilizing serial blood samples from 25 patients at C1D1 and C2D8, we observed a decrease in methylation in 22/25 (88%) patients ($P = 1.165 \times 10^{-6}$), providing a proof-of-mechanism. In the aforementioned clinical trial combining guadecitabine and pembrolizumab in refractory solid tumors, a significant reduction in LINE-1 DNA methylation following treatment, mostly pronounced in PBMCs samples at C2D8 (median 48.7%) compared with baseline (median 64.3 %).

7) The upregulation of CXCL9 and CXCL10 expression is quite interesting and should be explored further in RCC patients treated with these combinations. However, including studies on gene promoters would greatly help the interpretation of these results, and the mechanistic role of guadecitabine in this phenomenon.

We thank the reviewer for the comment and suggestion. We agree that the change in CXCL9 and CXCL10 expression is interesting and deserves further exploration. We added methylation data for a region of the CXCL10 promoter for a subset of the patients (top 15% and lower 15% changes in CXCL10). We went for this specific region of CXCL10 because it had been previously shown to be demethylated in a cell line. While we did see an overall decrease in methylation in the patients tested, PBMC cells did not show the same level of methylation as the cell lines at baseline (C1D1). CXCL10 (and CXCL9) promoter do not have a CpG island within their promoter region, making selection for methylation studies challenging. Literature also suggests that histone modifications are also involved in regulation of chemokine expression. As guadecitabine will demethylate in a whole genome manner, it would be most useful to perform a whole genome methylation assay using the Illumina MethylationEPIC arrays. That was outside the scope of this study.

Reviewer #2 (Remarks to the Author): with expertise in biostatistics, clinical trial study design

Thank you for giving the opportunity to review your manuscript. Please find below my comments/questions:

1. Why only doses of guadecitabine were chosen? How long was the total duration of the treatment schedule? I might have missed it in the manuscript, if it's there.

Thank you for the comments, this clarification was added in the paper, Guadecitabine SC at 60 mg/m² Daily×5 was tested in hematology clinical trials as single agent and was deemed safe. Issa JJ *Lancet Oncol.* 2015;16(9):1099-1110.

Same dose was utilized in 2 other phase 3 trials in hematologic malignancies: ClinicalTrials.gov Identifier: NCT02920008 and NCT02907359

We decided to start with this dose in combination with durvalumab and de-escalate to 45 mg/m² in case of DLT.

Treatment was given until disease progression or unacceptable toxicities.

2. How would you comment on the broad range age? Is this a limiting factor?

This study enrolled patients Age ≥ 18 years at the time of consent. No other limits on ages. We do not think this is a limiting factor.

3. Which is the list of toxicities that you would deem as dose limiting and therefore count towards an event in Phase Ib? You only mention grade 3 neutropenia as the only one of interest.

To eliminate confusion, we have decided to report phase II study only in this manuscript and reference our previously reported phase I results. Manuscript has been modified accordingly.

4. There is no justification as to the method used for the dose finding approach. You base your decisions on 6 patients, 3 on each dose. This is a very small number to declare a RP2D, how confident are you on this result?

We appreciate this comment and agree with it, however this was the original protocol design based on extensive studies of guadecitabine in hematologic malignancies that tested same doses and served as reference to our clinic trial. Issa JJ *Lancet Oncol.* 2015;16(9):1099-1110.

Same dose was utilized in 2 other phase 3 trials in hematologic malignancies: ClinicalTrials.gov Identifier: NCT02920008 and NCT02907359.

We have modified the manuscript to report phase II only and reference our previously published phase 1 results.

5. There is no description as to which summary statistics were used to describe data, no mention on the methodology or rules of the escalation/de-escalation study. In addition, there is no justification as to the sample size of Phase 1b.

We have modified the manuscript to report phase II only and referenced our previously published phase 1 results.

6. It'd be helpful to see summaries of prior treatment in Cohort 2 (duration, type of treatment). How was this accounted for in the analysis?

Cohort 2 was a small exploratory cohort of 15 patients, looking for signal finding in checkpoint inhibitor refractory setting, All patients received prior PD-1 inhibitor based therapy (7 patients nivolumab single agent, 5 patients with ipilimumab and nivolumab, 3 patients with prior pembrolizumab and axitinib). This was added to the paper.

No formal analysis was performed in that cohort.

Reviewer #3 (Remarks to the Author): with expertise in immunotherapy, renal cancer, clinical

Thank you for asking me to review the manuscript, "Phase Ib/II Study of Durvalumab and Guadecitabine in Advanced Clear Cell Renal Cell Carcinoma: Big Ten Cancer Research Consortium BTCRC-GU16-043". In this manuscript by Zakharia et al., the authors reported a safety and overall response data from a Phase Ib/II trial using durvalumab plus a hypomethylating agent (guadecitabine) in patients with metastatic clear cell RCC (mccRCC). They had 2 cohorts of patients. Cohort 1 included patients who were immune checkpoint therapy naïve (n=42) and cohort 2 included patients who were previously treated with immune checkpoint therapy (n=15). They reported that the combination is overall safe. In Phase II, an OR of 6% and PFS of 18.4 months were noted in cohort 1, while cohort 2 had an OR of 7% and PFS of 3.9 months. Further, they also reported immune monitoring studies on pre-treatment and post-treatment PBMC samples.

Overall, the manuscript has significant shortcomings both in the clinical and correlative analysis. My specific comments are as follows:

My specific comments are as follows:

Clinical:

1. Although, the combination is safe, OR and PFS rate was not impressive even in cohort I over the standard of care options. Therefore, utility of clinical translation is limited.

We thank the reviewer for the comment, and we acknowledge this observation, however we still think the results and biomarkers work merit the publication and would add to literature given the paucity of epigenetic modulations in solid tumors as opposed to hematologic malignancies.

2. Most of the patients (93%) were intermediate or poor risk. One would have expected a better response to immune checkpoint therapy than demonstrated in the trial.

We thank the reviewer for the comment, and we acknowledge that intermediate and poor risk patients tend to benefit better from immunotherapy ipilimumab and nivolumab combination when studies in first line setting in Checkmate 214. We think the small sample size could have

contributed to the lower response we encountered, nevertheless the PFS signal in cohort 1 is intriguing.

3. Demographics are not well matched. 77% of the patients were male. Age range is also very wide.

Thank you for this observation. This study enrolled patients Age \geq 18 years at the time of consent. No other limits on ages. The study did not put a limit on the male/ female ratio. We acknowledge that the smaller sample size and the nature of single arm study could have contributed to the imbalanced demographics.

4. In cohort 2, it was not clear whether patients received single agent immune checkpoint therapy, or combination of two immune checkpoint agents or immune checkpoint agent plus TKI. This information is important.

Cohort 2 was a small exploratory cohort of 15 patients, looking for signal finding in checkpoint inhibitor refractory setting, All patients received prior PD-1 inhibitor based therapy (7 patients nivolumab single agent, 5 patients with ipilimumab and nivolumab, 3 patients with prior clinical trial of pembrolizumab and axitinib). This was added to the paper.

5. Responses evaluation was not centralized but investigator assessed.

We agree with the comment, this investigator-initiated study was initially designed as an exploratory and signal finding of gadecitabine and durvalumab in metastatic renal cancer, as an attempt to position epigenetic modulation in solid tumors. The protocol did not require central review, which we agree is a limitation of our study.

6. Statistical methodology, study design is not properly explained in the manuscript.

This is explained in the Online Methods:

Study design: BTCRC-GU16-043 (NCT03308396) was a multicenter, single-arm phase Ib/II clinical trial in patients with metastatic ccRCC (pure or mixed histology) conducted throughout the Big Ten Cancer Research Consortium (Big Ten CRC). All participating sites obtained approval from their IRBs. The phase Ib portion evaluated the de-escalating doses of guadecitabine (level 0: 60 mg/m² and level -1: 45 mg/2 subcutaneously on Days 1-5) along with standard dose of durvalumab 1500 mg/m² on Day 8 every 4 weeks. The phase II part included patients in two cohorts: Cohort 1 (n=42) enrolled patients who were treatment naïve or with one prior line of therapy that is not CPI. Cohort 2 (n=15) enrolled patients with up to two prior systemic therapies including at least one CPI. All phase II patients were treated with the recommended phase 2 dose (RP2D) of guadecitabine along with durvalumab until disease progression or unacceptable toxicity or patient's withdrawal of informed consent. Treatment beyond first progression was permitted if patient was clinically stable and derived benefit with the study as per treating physician's discretion and provided additional informed consent.

Statistical methodology: We use binomial test to test our assumed ORR of 25% with durvalumab monotherapy. The binomial test is a statistical test used to determine if the proportion of successes in a binary outcome is significantly different from a particular value or expected proportion. It is based on the binomial distribution, which models the probability of getting a certain number of successes in a fixed number of independent trials.

Immune monitoring data:

Authors had 4 main figures reporting the immune monitoring data. However, the data lacks rigor in investigating the immunological correlates. The authors have used PBMCs for assessing the immunological correlates. Though this technique is less invasive, yet the peripheral immune landscape does not always reflect the intra-tumoral immune landscape. In that case, pre and post-treatment biopsies would provide critical relevant information.

We thank the reviewers for the comment, and agree with it, this study enrolled during COVID pandemic which limited the opportunities to get pre and post treatment biopsies due to hospitals restrictions on research procedures.

1. The authors have mentioned differences in abundance of MDSCs. However, more information is required on the type of MDSCs. What markers were used to define them?

MDSC were determined as myeloid cells CD45+Lin(-) CD33+ that either are not expressing monocytes/macrophage marker CD14 or are MHC-II negative. We previously showed that this MDSCs efficiently inhibited autologous T cell activation in human cancer setting (Immunity. 2013 Sep 19; 39(3), PMC3831370, Cancer Res. 2016 Jun 1;76(11):3156-65; PMID: 27197152). In the revised version, we extended the description of the gating in the materials and methods section and in the legend for individual figures.

2. Authors have mentioned that TNFa was higher in all lymphocyte subsets (Figures 2D-G). But data is shown only for T cells and ILCs.

The reviewer is right. We analyzed TNF expression only in T cells and ILCs. We now corrected the text accordingly.

3. Rather than saying higher expression of FoxP3 in T cells, better to show the abundance of these cell types? Do T cell suppression assays to assess Treg phenotype?

Unfortunately, immunotherapy induces expression of CD25 on effector T cells and limits the possibility of using CD25 to isolate Treg cells from patients' blood. However, we previously

comprehensively validated FOXP3 as a marker to determine primary Tregs in cancer patients (Cancer Res. 2009 1;69(9) PMID: 19383912).

4. RORgt in CD4 is TH17. What is its impact in CD8 T cells and their ability to mount anti-tumor immune response?

Added to discussion: CD8+ T cells expressing ROR γ t (so-called TC17 cells) have been shown to promote inflammation, contribute to defense against infections, and participate in autoimmunity (reviewed in PMID: 27173097). On the other hand, RORgt activates stemness pathways in T cells keeping Th17 cells in a less-differentiated state capable of superior persistence and functionality in mice (Immunity. 2011 PMID: 22177921) and human (Sci Transl Med. 2011 Oct 12;3(104), PMC3345568) models. This property may be critically important for controlling also CD8 T cell's response to neoantigens (Cell. 2021 Sep 16;184(19), PMID: 34534464). Definitely this subject is worth id further investigations.

5. Authors have mentioned that certain immune parameters are “correlated” with response/toxicity. However no statistical tests have been done to measure correlation and no R values have been given.

We used the Anova method to test differences in immune cell phenotype between groups with a particular type of clinical response. While relationship between toxicity and peripheral immune status was tested by both Anova or T-test). We agree that our original version lacked a proper description of the statistical methods. In the current version, we have added statistical details to the figure descriptions and the materials and methods section.

We agree with the reviewer that using term “correlation” for relationships tested by Anova may be misleading and have corrected the text accordingly.

6. Instead of looking only at cxcl9/10, maybe the authors can consider doing a luminex panel/s? Also flow is not the best way to look at chemokine expression.

The reviewer is right. We regret that we did not include information that chemokines were measured with a Luminex/multiplex. In the current version this information is included in the description of the figure's legend. We include data from whole chemokines panel in Figure 5 and Supplementary S4.

7. Fig. 4c, d are main figures but there is no statistical analysis was reported.

Figure 4 b, c, d (Currently Fig. 5 b,c,d) shows the Gardner-Altman plot for MIG/CXCL9, IP-10/CXCL10 and I-TAC/CXCL11. In each case, two dose points are considered C1D1 and C2D8 which are shown on the x-axis. For each patient, we show the increase or decrease in levels based on a cut-off with blue and orange lines respectively. We performed a paired t-test to find out if there is a significant difference in dose level between two dose points and p-values are 0.00007, 0.00001 and 0.000007 respectively. In all cases the p-values are less than the standard cut-off 0.05.

8. Checking hypomethylation in specific genetic regions which have been mentioned in the manuscript via bisulfite genomic seq so that the change in methylation can be correlated with the immune parameters discussed (for eg, increase in cxcl9/10).

We added amplicon bisulfite sequencing for a region of the CXCL10 promoter for a subset of the patients, i.e those with changes of at least 15% (up or down) in CXCL10 expression.

9. On what basis was the 15% cutoff selected for binomial gating of patient samples?

The 15% cut-off was determined arbitrarily by testing significance in 15%-steps of 15%, 30%, and 45%, respectively.

10. The authors have discussed that the increase in cxcl9/10 could serve as a biomarker to “predict” responders from non-responders? However, the authors have not shown any correlation of response with pre-treatment baseline values of cxcl9/10. Then can this be considered a “predictive” biomarker?

We agree with reviewer that based on the binomial proportion test, the increase in CXCL9, CXCL10, and CXCL11 was observed with best RECIST response of (SD, PR and CR) group (P = 0.054 and P=0.000208, and 0.000208 respectively) as illustrated in table S2 and a trend towards improvement in PFS mainly with CXCL 10, 11. This observation is thought provoking, however given the small sample size, we are hesitant to call it as “predictive biomarker”. We believe this need to be validated in larger studies.

In regard to the correlation with pre-treatment baseline value of CXCL 9, 10 and 11. The table below shows the increase and decrease in dose level for CXCL11, 10 and 9 with baseline (C1D1) only. We considered the log-scaled values to control the wide range and subtracted the minimum values followed by a multiplication 100/maximum to obtain percentages on the same scale. We use a 15% cut-off to determine the patients with increase and decrease levels. In this case, we observe that most of the patients in CXCL10 and CXCL9 are related to the decreased baseline level. (p-values are 0.00004 and 0.000007 respectively). This did not hold true for CXCL11. Given the small sample size, we elected not to include this part in the manuscript, however, if the reviewer finds a value in adding this part, we will be happy to incorporate to the supplements.

Characteristic	CXCL 11		CXCL 10		CXCL 9	
	Increase	Decrease	Increase	Decrease	Increase	Decrease
Best RECIST Response						
CR	1	0	1	0	1	0
PD	7	0	1	6	4	3

PR	4	1	1	4	2	3
SD	16	3	3	16	3	16
Total	28	4	6	26	10	22

11. In all Figures, please draw lines to indicate which groups are being considered for statistical tests.

This was included in the updated figures.

12. Include details of the statistical tests done to investigate differences in immunological correlates.

This was added to the statistical analysis.

13. Is Data shown as Mean+/-SEM or SD?

For figure 5 b, c and d, we only show the data points. Our focus is to find out how significantly different the pairwise mean differences from zero rather than an overall average or any other dispersion measures.

14. All Fig. 2 panels have been wrongly referred to as Figure 1 in the legend.

We corrected all wrong referrals in the text.

15. In Fig. S1a- no live/dead gating.

This was addressed in the figure.

16. In Fig. S1a- no specific markers (for eg. CD19) used to define B cells.

B cells were gated as CD45⁺CD15⁻CD19⁺ (corrected Figure S1)

17. Fig. S1a- Specific markers for ILCs.

ILCs (gated as CD45⁺CD19⁻CD3⁻CD7⁺ (corrected Figure S1)

18. Table S2- CXCL9/10 changes were in the same patients or different patients?

They were tested in the same patients.

19. Table S2- SD, PR and CR have been grouped together, instead group SD with PD?

We considered SD lasting longer than 6 months as meaningful benefit to patients, hence it was grouped with PR/ CR.

Reviewer #4 (Remarks to the Author): with expertise in immunotherapy, renal cancer, clinical

Phase 1/2 study of PDL1i + hypomethylating agent in advanced RCC. The combination of Durvalumab + guadecitabine proved to be too toxic at starting dose, and the ph2 piece of the trial did not meet its efficacy goal. An in-depth peripheral-blood based biomarker analysis was conducted (mostly to address heterogeneity in peripheral immune composition / activation of various compartments; less so around the interplay between methylation and immune activation) and is included with several findings of interest.

- Protocol states dose escalation plan (section 5.3 protocol): “• Six patients will be enrolled at dose level 0. If 2 or fewer patients experience a dose limiting toxicity, the study will continue to the phase II portion. • Alternately, if 3 or more patients have a dose limiting toxicity at dose level 0, 6 patients will be accrued at the lower dose (dose -1). If 2 or fewer patients experience a dose limiting toxicity, the study will continue to phase II at dose level -1..” The results section of the paper states, 3 + 3 patients were enrolled at 60mg (level 0) and 45mg/m² (level -1) with one 1 DLT seen. The protocol section 5.5 “Treatment plan for the Phase II Portion” states “three of five Phase Ib subjects who received dose of 60mg/m²...”

It is unclear how many pts were treated on ph1b portion (3 vs 5) and why it wasn't a total of 6, per the protocol?

We thank the reviewer for the comment, total of 6 patients enrolled in phase 1, and we previously reported that. <https://doi.org/10.1016/j.annonc.2020.08.844>.

To eliminate confusion with many cohorts, we have decided to report phase II study only in this manuscript and reference our previously reported phase I results. Manuscript has been modified accordingly.

- Table 1 / results: further information should be provided regarding exposure to prior therapies (median # lines at the least) to help put findings in perspective

In Cohort 1: 15/36 (42%) received 1 prior line of therapy. All the rest of subjects (58%) were treatment naïve.

In cohort 2: all 15 subjects (100%) received a prior line of therapy. Seven patients treated with nivolumab single agent, 5 patients with ipilimumab and nivolumab, 3 patients treated with pembrolizumab and axitinib

This was updated in the paper.

- The authors should clearly state which imaging exact assessment methodology was used on the trial – are responses reported here per RECIST 1.1, modified RECIST, iREcIST, etc.? “disease control rate” should be clearly defined (looks like they are proposing CR+PR+SD>6mo).

We Thank the reviewer for this clarification. This was clarified in the manuscript in the method section.

Response was assessed per RECIST v1.1 criteria.

Response by immune-related RECIST criteria was evaluated as a secondary endpoint.

Disease control rate (DCR) was defined in the manuscript as CR+PR+SD>6 months.

- It is not meaningful to report a median OS for cohort 1 (6 patients). Landmark OS more helpful; could consider reporting that for all patients – rather than separate for ph1b and ph2

Thank you for the comment, we agree thy way we reported the initial manuscript could create some confusion. Hence to better clarify, we decided to report phase II only and reference our previously reported phase 1 study.

The median OS reported for cohort 1 in the manuscript represent 36 patients treated in the phase II part of the study who are checkpoint inhibitors naïve.

- Adverse events: attributions were per treating investigators? Please state clearly; table 2: are these treatment-emergent or treatment-related?

This is added to the paper, adverse events were reported by the treating investigators and were graded per Common Terminology Criteria for Adverse Events (CTCAE; version 4.03).

Table 2 represent all emergent adverse events while on study, this was clarified in the table section.

- IRAE – acronym not defined in text

Immune related adverse events (IRAE) was added to the manuscript.

- Adverse events: the ph1b findings should be reported by dose level (separate table). With that, a more detailed explanation is needed for the DSMC’s recommendation to pursue DL-1 as RP2D –Safety results section states ‘to avoid dose delays for neutropenia’; Table 2 notes G4 neutropenia in 39% of the cohort , but the results section does not highlight this appropriately (G4 not mentioned in the text at all; table 2 does not break things down by cohort). How soon did neutropenia set in? were patients dose-reduced/rechallenged? If yes, how did they do? all these would be relevant to better understand this key issue with the combination.

We Thank the reviewer for this comment, and we agree the way we reported initially is rather confusing. To eliminate confusion, we have decided to report phase II study only in this manuscript and reference our previously reported phase I results. Manuscript has been modified accordingly.

- legends figures 2 and 3 each contain an error (reference figure 1 and 2, respectively; when it should be 2 and 3, respectively).

Thank you for bringing this obvious inaccuracy to our attention. We corrected all wrong referrals in the text.

- Flow cytometry ("Effect of different...") text reference significant correlation (MDSC) or the lack thereof; p-values provided in figures 2 and 3. the text (and fig legend) should reference which statistical test was used to test for association (not just naked p value). I cannot find this in the methods section either.

We regret incomplete description of statistical methods. Appropriate annotations have been placed in the Figure legend and material and methods section.

- Biomarker findings: for the most part these are testing hypotheses around the IO piece of the regimen; inverse efficacy correlation with myeloid populations is not a novel finding, nor is the interferon-gamma signal

There are few reports on the relevance of MDSC and IFN- γ signal measurements on the baseline level in the blood of patients with mostly melanoma and prostate cancer for their response to neoadjuvant immunotherapies. However, even ccRCC is considered an immunogenic tumor, it is relatively resistant to immunotherapy. Recent trials proposed alternative combination therapies to overcome resistance, but they haven't offered any reliable biomarkers. The most studied biomarker PD-L1 failed to demonstrate a predictive capability in metastatic RCC. Therefore, any correlation that may point to a response biomarker, even if it has been shown for other cancers, is extremely relevant to RCC treatment.

- Biomarkers and immune-mediated toxicity: the manuscript should provide a dedicated table breaking down incidence and types of irAEs (vs. table2 which is all AEs)

Another table (table 3) for immune related adverse events (irAE) is added.

- T-bet baseline / on treatment; this is an interesting finding; which cells were tested for expression? were there patients who did not express T-bet at baseline but did so at a later timepoint, and did they suffer high-grade AEs?

We tested T-bet in CD8+ T cells (Fig. 3F) CD4+ T cells (Fig. 3G), and ILCs (Fig. 3H). As can be seen from these figures, all patients expressed T-bet, with positivity varying by cell type, 70% ILCs, 45% CD8+ T cells, and 18% CD4 T cells. We did not notice any significant differences in treatment time in CD4+ T cells and ILCs subset (maybe due to too few patients). As shown in Fig. 3I patients with high-grade AEs showed an increase in T-bet in CD8+ T cells, while this

expression in patients without AEs tended to decrease. This observation, however, requires further research.

- CXCL9&10 levels in peripheral blood: this is one of the most relevant correlative analyses for the combination, and it is interesting to see the trends highlighted in figure 4. what was the proportion of patients with increase vs. decrease? The text (and figures, ideally) should state. The No at risk #s in 4C suggest 7 increase 25 decrease? How many w/o changes? Why are they not included in Fig 4C/D?

Table S2 summarizes the number of patients with increase/decrease dose level for CXCL11, CXL10 and CXCL9 respectively along with their corresponding best responses.

Characteristic	CXCL 11		CXCL 10		CXCL 9	
	Increase	Decrease	Increase	Decrease	Increase	Decrease
Best RECIST Response						
CR	0	1	0	1	0	1
PD	2	5	2	4	2	4
PR	2	3	3	2	1	4
SD	9	10	8	12	4	16
Total	13	19	13	19	7	25

- Figures 4C and D: would be more interesting to see increase/decrease curves in 1 graph and visualize the comparison

Fig 4c, d are changed to Fig 5 b, c, d in the current version. We have provided all the combined figures in S4 in the new version.

- Pyrosequencing LINE-1: this is the 2nd biomarker analysis of specific interest for the COMBINATIOn, i.e. of central importance to the paper. Considering this, the section is rather short (vs flowcytometry of relevance to durva which in turn is not novel and could be shorter...). How do changes in CpG methylation correlate with efficacy? with neutropenia? the authors should provide further detail.

We thank the reviewer for this comment, we agree with the importance of correlating CpG methylation with efficacy and neutropenia, however in discussion with statistician we thought with the small cohort of 12 patients tested, it would be difficult to draw a solid conclusion. Certainly, this is a great question for future studies.

- Were genomics assessed and is there correlation between alteration in recurrent events (histone-regulating genes?) and response?

Unfortunately, this was not performed as part of the study.

REVIEWERS' COMMENTS

Reviewer #2 (Remarks to the Author):

Many Thanks for addressing my comments in detail.

No further comments from me.

Reviewer #3 (Remarks to the Author):

Thank you for asking me to review the revised version of the manuscript "Phase Ib/II Study of Durvalumab and Guadecitabine in Advanced Clear Cell Renal Cell Carcinoma: Big Ten Cancer Research Consortium BTCRC-GU16-043". I appreciate the authors for providing a more detailed description of the statistical plan in the revised version. They also provided additional luminex data. However, the concerns regarding the trial design and clinical relevance of the data still remains. Further, the role of CXCL9,10 and 11 has been described in the literature before. The correlative analyses lack depth and insight.

Reviewer #4 (Remarks to the Author):

all of my prior comments were addressed

Reviewer #2 (Remarks to the Author):

Many Thanks for addressing my comments in detail.

No further comments from me.

Thank you for your time and feedback!

Reviewer #3 (Remarks to the Author):

Thank you for asking me to review the revised version of the manuscript "Phase Ib/II Study of Durvalumab and Guadecitabine in Advanced Clear Cell Renal Cell Carcinoma: Big Ten Cancer Research Consortium BTCRC-GU16-043". I appreciate the authors for providing a more detailed description of the statistical plan in the revised version. They also provided additional luminex data. However, the concerns regarding the trial design and clinical relevance of the data still remains. Further, the role of CXCL9,10 and 11 has been described in the literature before. The correlative analyses lack depth and insight.

We thank Reviewer 3 for the time and feedback!

We agree that the importance of Th1 chemokines in anti-tumor immunity has been reported. Previously we have shown that DMT1-dependent epigenetic silencing of CXCL9 and CXCL10 determines effector T-cell trafficking to the tumor microenvironment and affects the therapeutic efficacy of checkpoint blockade treatment (Peng D. et al. Epigenetic silencing of TH1-type chemokines shapes tumor immunity and immunotherapy Nature. 2015 Nov 12;527). The fact that guadecitabine in combination with durvalumab affects the expression of Th1 chemokine and improves peripheral immune markers in clinical practice is, in our opinion, extremely important. This indicates that this therapeutic strategy can effectively break the tumor-related epigenetic block. This provides a rational basis for future randomized clinical trial. We agree with the reviewer that the results are correlational. Unfortunately, this is a common problem with research based on human material, which, in our opinion, does not diminish their value.

Reviewer #4 (Remarks to the Author):

all of my prior comments were addressed.

Thank you for your time and feedback!